# In Vivo Two-Photon Imaging Analysis of Dynamic Degradation of Hepatic Lipid Droplets in MS-275-Treated Mouse Liver

**DOI:** 10.3390/ijms23179978

**Published:** 2022-09-01

**Authors:** Chang-Gun Lee, Soo-Jin Lee, Seokho Park, Sung-E Choi, Min-Woo Song, Hyo Won Lee, Hae Jin Kim, Yup Kang, Kwan Woo Lee, Hwan Myung Kim, Jong-Young Kwak, In-Jeong Lee, Ja Young Jeon

**Affiliations:** 1Department of Medical Genetics, Ajou University School of Medicine, Suwon 16499, Gyeonggi-do, Korea; 2Three-Dimensional Immune System Imaging Core Facility, Ajou University, Suwon 16499, Gyeonggi-do, Korea; 3Department of Physiology, Ajou University School of Medicine, Suwon 16499, Gyeonggi-do, Korea; 4Department of Biomedical Science, The Graduate School, Ajou University, Suwon 16499, Gyeonggi-do, Korea; 5Department of Endocrinology and Metabolism, Ajou University School of Medicine, Suwon 16499, Gyeonggi-do, Korea; 6Department of Energy Systems Research, Ajou University, Suwon 16499, Gyeonggi-do, Korea; 7Department of Chemistry, Ajou University, Suwon 16499, Gyeonggi-do, Korea; 8Department of Pharmacology, Ajou University School of Medicine, Suwon 16499, Gyeonggi-do, Korea

**Keywords:** MS-275, NAFLD, fatty liver, intravital imaging, FGF21, mitochondrial oxidation

## Abstract

The accumulation of hepatic lipid droplets (LDs) is a hallmark of non-alcoholic fatty liver disease (NAFLD). Appropriate degradation of hepatic LDs and oxidation of complete free fatty acids (FFAs) are important for preventing the development of NAFLD. Histone deacetylase (HDAC) is involved in the impaired lipid metabolism seen in high-fat diet (HFD)-induced obese mice. Here, we evaluated the effect of MS-275, an inhibitor of HDAC1/3, on the degradation of hepatic LDs and FFA oxidation in HFD-induced NAFLD mice. To assess the dynamic degradation of hepatic LDs and FFA oxidation in fatty livers of MS-275-treated HFD C57BL/6J mice, an intravital two-photon imaging system was used and biochemical analysis was performed. The MS-275 improved hepatic metabolic alterations in HFD-induced fatty liver by increasing the dynamic degradation of hepatic LDs and the interaction between LDs and lysozyme in the fatty liver. Numerous peri-droplet mitochondria, lipolysis, and lipophagy were observed in the MS-275-treated mouse fatty liver. Biochemical analysis revealed that the lipolysis and autophagy pathways were activated in MS-275 treated mouse liver. In addition, MS-275 reduced the de novo lipogenesis, but increased the mitochondrial oxidation and the expression levels of oxidation-related genes, such as PPARa, MCAD, CPT1b, and FGF21. Taken together, these results suggest that MS-275 stimulates the degradation of hepatic LDs and mitochondrial free fatty acid oxidation, thus protecting against HFD-induced NAFLD.

## 1. Introduction

Non-alcoholic fatty liver disease (NAFLD), a chronic disease with a rapidly increasing incidence worldwide, is characterized by steatosis and steatohepatitis, eventually resulting in fibrosis and cirrhosis [1]. Aspects of lipid metabolism, such as lipid uptake, synthesis, and storage, and the use of free fatty acids (FFAs) and triglyceride (TG) are tightly regulated in the normal physiological state. However, in NAFLD, hepatic steatosis is induced by an abnormal accumulation of lipid droplets (LDs) due to an imbalance between the lipid storage and use [2]. The lipid accumulation in hepatocytes involves the conversion of excessive FFA entering through the portal vein into TG, increased de novo lipogenesis, decreased mitochondrial fatty acid (FA) oxidation, and a reduction in the very-low-density lipoprotein (VLDL) secretion from the liver into the blood [3]. During lipid accumulation, incomplete FFA oxidation and lipid intermediates occur, leading to inflammation, ER stress, and insulin resistance, which exacerbates the pathology of NAFLD. Therefore, the degradation of hepatic LDs and complete FFA oxidation have become therapeutic targets for NAFLD.

Histone deacetylase (HDAC), an epigenetic regulator, is involved in lipid metabolism [4]. Whereas some of the HDACs contribute to lipid dysregulation, other isoforms play beneficial roles. HDAC6 binds with p62/SQSTM1 (sequestosome1) and promotes the breakdown of LDs by inducing selective autophagy in the hepatocyte-like oenocytes in response to starvation [5]. HDAC6 can regulate the acetylation of the lipid-binding protein cell death-inducing DFFA-like effector c (CIDEC), increase the fat accumulation in adipocytes, and reduce insulin sensitivity [6]. However, MS-275, an inhibitor of HDAC1/3, protects against lipotoxicity in C2C12 myotubes, and ameliorates hyperglycemia and insulin resistance in high fat/high fructose-treated mice [7]. It can also reduce fat mass and adipocyte size by enhancing oxidative metabolism in the skeletal muscle and adipose tissue of the high-fat diet (HFD)-induced obese mice and db/db mice [8,9,10]. In addition, He et al. demonstrated that butyric acid (BA) and its derivative, 4-phenylbutyric acid (PBA), downregulated the de novo lipogenesis by inactivating the SREBP1-mediated transcription of FASN, SCD-1, and ACC, and promoting hepatic β-oxidation [11]. The MS-275 treatment exerts a beneficial effect on NAFLD and metabolic diseases, such as obesity and type 2 diabetes mellitus (DM) [12]. However, how MS-275 breaks down fat droplets, interacts with mitochondria, and increases β-oxidation in fatty liver disease remains unclear.

The intravital imaging of the catabolism and anabolism of LDs in living animals can provide insight into the mechanisms of NAFLD and the effects of drug treatment in animal models [13]. The sizes of the individual hepatic LDs in hepatocytes gradually increase from 3 to 10 µm over 21 days in methionine- and choline-deficient (MCD) mice, whereas the LDs in the livers of mice fed a normal chow diet (CD) are <1 µm in diameter [13]. As the fat droplets grow, the area and length of the sinusoidal vessels decrease. The vessels are deformed due to rarefaction and dilation in the liver of the streptozotocin (STZ)-treated HFD-fed mice compared to the CD-fed mice, which show a uniform distribution of dense sinusoidal vessels [14]. Using in vivo imaging, Diniz et al. demonstrated that the population of liver leukocytes I was altered at the early stage of hepatic steatosis, comprising >50% dendritic cells (DCs) and Kupffer cells (KCs) [15]. The number of neutrophils infiltrating the liver is reduced in middle-stage NAFLD [16]. During NAFLD, a dramatically changed hepatic immune milieu is presumed to occur in response to a fatty liver environment [17]. However, the role of each type of immune cell is unclear.

The intravital imaging studies on the dynamic degradation of LDs have examined HFD-induced fatty liver [18]. For the first time, this study investigated the effects of MS-275 on the degradation of LDs using in vivo intravital images of HFD-induced fatty liver. As HFD progressed in mice, tiny hepatic LDs interacted, fused to large LDs, and accumulated in hepatocytes. However, short-term treatment with MS-275 increased the interaction between LD and the lysozymes in HFD-induced fatty liver, which reduced the number of the large LDs but increased that of the tiny LDs. Electron microscopy (EM) analysis revealed that lipolysis, lipophagy, and many peri-droplet mitochondria (PDM) were observed in the MS-275-treated fatty liver. The biochemical analysis demonstrated that activation of the lipolysis pathway (AMPK, PKA, and p-CREB) and autophagy pathway was evoked in the MS-275-treated mouse liver. Finally, MS-275 reduced de novo lipogenesis, but increased mitochondrial oxidation and the expression levels of oxidation-related genes, such as *Pparα*, *Acadm*, *Cpt1b*, and *Fgf21*. We report that MS-275 in fatty liver degraded LDs and promoted mitochondrial FFA oxidation. It is important to understand the mechanism of catabolism and anabolism of LDs for the development of drugs for NAFLD.

## 2. Results

### 2.1. Formation of Hepatic LDs in HFD-Induced Fatty Liver

In obesity, an imbalance between lipid synthesis and degradation in the liver can lead to hepatic steatosis [19]. HFD-fed mice develop hepatic steatosis via multiple metabolic pathways including the uptake of a large amount of FFAs by hepatocytes, conversion thereof into TGs to avoid fatty acid oxidation (FAO), and reductions in VLDL and lipophagy [19]. Finally, hepatocytes store synthesized TGs in the cytoplasm as LDs via a complex and dynamic process [20]. To investigate the formation of LDs during the development of hepatic steatosis due to a HFD, we devised an intravital imaging system. The mice were anesthetized and intravenously injected with LD1 red fluorescent dye. Next, the liver was surgically exposed in a heating chamber, fixed on a Sylgard pad, covered with a coverslip, and observed using a multiphoton microscope (Figure 1A). The LD formation was observed by 3D rendering of the intravital images at 8 and 14 weeks after feeding a HFD (Figure 1B). In the multiphoton microscopy images, the number and size of LDs were markedly increased by a HFD (Figure 1C). The total volume of LDs increased as the hepatic steatosis progressed. In particular, during the hepatic steatosis progression, large LDs (>10,000 µm^3^) were increased at 14 weeks after feeding the HFD compared to 8 weeks. However, medium-sized (1000–10,000 µm^3^) and small (<1000 µm^3^) LDs decreased slightly (Figure 1D). As shown in Figure 1E, the hepatic steatosis progression due to a HFD involved the fusion of small LDs to large ones Appendix A. Western blot analysis showed that administration of an HFD upregulated the class 1 HDAC proteins, such as HDAC1 and 2 (Figure 1F). These results suggest that a HFD promotes the formation of large LDs via the fusion of small and medium-sized LDs, and elevates the expression levels of class 1 HDACs.

### 2.2. MS-275 Ameliorated Fatty Liver Disease in a Mouse Model of NAFLD

MS-275, a HDAC 1/3 inhibitor, has a beneficial effect on obesity, insulin resistance, and diabetes [12]. Therefore, we investigated the effect of MS-275 on NAFLD. To induce liver steatosis, C57BL/6J mice were fed a HFD for 14 weeks and mice with fatty liver were intraperitoneally injected with MS-275 (10 mg/kg), five times every other day (Figure 2A). The livers of the HFD-fed mice were significantly heavier than those of the CD-fed mice. However, liver weights after MS-275 treatment were significantly reduced compared to those of the HFD-fed mice (Figure 2B). In accordance with the increased amount of LDs, the liver TG level was significantly increased in the HFD group compared to the CD group. However, MS-275 treatment significantly decreased the amount of LDs and the liver TG level (Figure 2C). Hematoxylin and eosin (H&E) and oil red O (ORO) staining revealed dramatically increased lipid vacuoles and an abundance of red LDs, which were also increased in size, in the liver tissues of the HFD-fed mice (Figure 2D,E). However, MS-275 decreased the amount of lipid vacuoles and LDs in the liver. In addition, the serum levels of TG, total cholesterol (TCHO), aspartate aminotransferase (AST), and alanine-aminotransferase (ALT) were increased by the HFD, but significantly reduced by the MS-275 treatment (Figure 2F). These results suggest that MS-275 attenuated HFD-induced hepatic steatosis and liver injury.

### 2.3. MS-275 Stimulated Dynamic Degradation of Hepatic LDs in Mouse Fatty Liver

To determine the mechanism by which MS-275 reduced the LDs in fatty liver, we evaluated the pattern of reduction in the LDs by MS-275. To visualize the LDs and intrahepatic blood vessels, LD1 was administrated intravenously into the MS-275 injected HFD-mice and LD formation was observed by multiphoton microscopy. As shown in Figure 3A, the MS-275-treated mouse liver had smaller LDs than the HFD-fed mouse liver (Figure 3A). The volumes of LDs were significantly decreased by the number of injection times with MS-275. Interestingly, the LDs > 10,000 µm^3^ were decreased, whereas the medium-sized LDs (1000–10,000 µm^3^) were slightly increased, after three injections of MS-275. The hepatic LDs in livers were dramatically decreased after five injections of MS-275; only <1000 µm^3^ LDs remained (Figure 3C). In addition, liver sinusoidal microvasculature visualized in vivo by fluorescent labeling showed that the vessel diameter of the mice fed a HFD was markedly decreased, however the MS-275 increased the vessel diameter (Figure 3D). These results indicate that hepatic steatosis was improved by MS-275 via breakdown of the large LDs into smaller ones and improvement of liver sinusoidal microvascular structures.

### 2.4. MS-275 Increased Lipophagy- and Lipolysis-Related Gene Expression in Mouse Fatty Liver

We next evaluated which organelles were affected by MS-275. Since the degradation of the LDs is mediated by lysosomal activity [21], we investigated whether fusion of the LDs with lysosomes induces lipolysis in the liver tissues of MS-275-treated mice. Mice fed a HFD were treated with MS-275 and the lysosomes were labeled with two-photon probes for lysosome (BLT). Compared to HFD mice, the number of lysosomes near the LDs was significantly increased by MS-275 treatment (Figure 4A; Appendix A). Transmission electron microscopy (TEM) showed that the large LDs in the liver tissue of mice fed a HFD were surrounded by a soft and gentle membrane, whereas the surface of the small LDs from liver tissues of MS-275-treated mice was rough with small autophagosomes (Figure 4B). Western blotting showed that MS-275 stimulated the conversion of LC3I into LC3II, and p62 degradation (Figure 4C). The MS-275 also increased the mRNA levels of hepatic TG lipase (Lipc) and adipose TG lipase (Pnpla2) in the liver (Figure 4D). Interestingly, MS-275 induced the expression of adenine phosphoribosyltransferase (Aprt), which is implicated in adenosine monophosphate (AMP) biosynthesis, thereby activating AMPK, which is a master regulator of metabolism (Figure 4E). These results suggest that MS-275 enhances lipophagy and lipolysis-related gene expression by activating AMPK in mouse fatty liver.

### 2.5. MS-275 Increased Mitochondrial Biogenesis and Content, and Peri Lipid-Droplet Mitochondria

Since the hepatic LDs are reduced by decreased de novo lipogenesis, excessive FAO, increased VLDL, and excessive lipophagy [22], we examined which organelles were involved in the MS-275-mediated LDs degradation. Interestingly, a large number of LDs were in contact with mitochondria in the MS-275-treated livers (Figure 5A). The peri-droplet mitochondria (PDM) were increased in the MS-275-treated mice, particularly the small and large LDs. We therefore quantified the mitochondria in contact with LDs. First, we drew a red line around the mitochondria to determine their number and circumference. We then drew a blue line that contacted the site of mitochondria and LDs. The abundance of PDM was increased by MS-275 (Figure 5B). The number of mitochondria in contact with LD, mitochondria-LD in contact with the mitochondria perimeter, and mitochondria-LD in contact with the LD perimeter were increased by MS-275 (Figure 5C–E). In addition, MS-275 increased the mitochondrial content per unit area (Figure 5F), and the mRNA expression levels of transcriptional coactivation in the context of mitochondrial metabolism and biogenesis, such as mitochondrial transcription factor A (Tfam), nuclear factor erythroid derived-like 2 (Nrf2), and peroxisome proliferator-activated receptor gamma coactivator 1-alpha (Ppargc1a), were increased (Figure 5G). These data suggest that MS-275 enhances mitochondrial biogenesis, content, and the proportion of PDMs, resulting in degradation of LDs.

### 2.6. MS-275 Increased Mitochondrial Oxidation and FGF21 Expression

To elucidate the mechanisms underlying the effect of MS-275 on HFD-induced steatosis, we investigated the influence of MS-275 on mitochondrial FFA oxidation in primary mouse hepatocytes (PMH), using an FFA oxidation assay kit. The PMH pretreated with MS-275 for 16 h had an increased mitochondrial oxygen consumption rate (OCR) compared to the DMSO-treated control PMH. The area under the curve (AUC) of OCR was increased compared to the control (Figure 6A). Because MS-275 induced mitochondrial oxidation, we investigated the expression levels of the genes related to FAO and lipid synthesis. The HFD-induced fatty liver showed an increased expression of lipid synthesis-related genes, such as sterol regulatory element-binding protein (Srebf), fatty acid synthase (Fasn), diacylglyceride acyl transferase (Dgat), stearoyl-CoA desaturase (Scd), and acetyl-CoA carboxylase alpha (Acaca), whereas the expression of the FAO-related genes, such as proliferator-activated receptor alpha (PPARa), medium-chain acyl-coenzyme A dehydrogenase (Acadm), and carnitine palmitoyl transferase I b (Cpt1b) was unchanged (Figure 6B,C). The MS-275-treated mouse livers showed increased mRNA levels of FAO-related genes, but decreased mRNA levels of lipogenic genes (Figure 6B,C). In addition, several studies have reported that FGF21 overexpression influences weight loss and hepatic steatosis [23,24]. Therefore, we investigated the expression of Fgf21 in the livers of the MS-275-treated mice. The mRNA and protein levels of Fgf21 were increased in the MS-275-treated mouse livers (Figure 6D,E). The serum Fgf21 level was significantly increased by MS-275 treatment (Figure 6F). These results suggest that MS-275 can increase mitochondrial oxidation by upregulating the FAO-related genes and FGF21 in fatty liver.

## 3. Discussion

An abnormal accumulation of LDs in hepatocytes is a hallmark of NAFLD [25]. Abnormal fat synthesis and metabolism can lead to the accumulation of various fat metabolites [26]. NAFLD can progress to nonalcoholic steatohepatitis (NASH) via inflammation and fibrosis [27]. Bio-imaging data suggest that, as fatty liver progresses in mice, tiny individual hepatic LDs interact and fuse to form large LDs [28]. During the fatty liver progression, the area and length of sinusoidal vessels are decreased [14] and nonparenchymal cells, including infiltrated leucocytes, are pathologically changed [15]. However, there are a few imaging studies of lipolysis reporting that LDs interact with lysosomes and decrease in size after drug treatment. Through intravital imaging of the liver fat accumulation, we found that MS-275 decreased LD size and improved hepatic steatosis. The intravital imaging and TEM showed that MS-275 increased the interactions of LDs with lysosomes, and mitochondria. It also activated AMPK, inducing lipolysis by upregulating the expression of the hepatic lipolysis-related enzymes (lipase C, ATGL, and HSL). This resulted in reduced and elevated numbers of large and small LDs, respectively. In addition, MS-275 reduced de novo lipogenesis, but increased lipid oxidation and mitochondrial biogenesis. Recent studies have revealed that ceramide contributes to de novo lipogenesis, pathogenesis of NAFLD, and impaired mitochondrial fatty acid oxidation [29,30]. In our study, the treatment with MS-275 decreased the mRNA expression levels of fatty acid synthase, which is known to regulate palmitoyl–CoA expression, and it might be predicted to inhibit ceramide production [31]. Therefore, it would be considered that inhibiting ceramide production and toxicity by MS-275 might prevent NAFLD. However, treatment with MS-275 had no significant effect on the ceramide generation in cancer cells [32]; more detailed further investigation is required to determine whether MS-275 affects the ceramide production in metabolic disease states, such as NAFLD.

HDACs disturb energy metabolism by regulating epigenome modifiers [33]. Inhibition of HDAC can improve metabolically disturbed adipose, liver, muscle, and cardiac tissue [9]. MS-275, a class I-specific HDAC inhibitor, primarily inhibits HDAC1–3 [12]. MS-275 enhances oxidative and energy metabolism in adipose tissues. The selective ablation of HDAC3 in white adipose tissue (WAT) enhanced the oxidative capacity and led to WAT browning [34]. MS-275 improved the glycemic control and reduced obesity in diet-induced obese mice by augmenting the glucagon-like peptide-1 receptor in pancreatic beta cells [12]. In this study, the injection of HFD-induced obese mice with MS-275 prevented hepatic steatosis by reducing de novo lipogenesis and increasing lipolysis and mitochondrial oxidation, by increasing the expression of oxidation-related genes, such as PPARα, MCAD, CPT1b, and FGF21. The liver-specific deletion of HDAC3 induces lipid synthesis and storage within LDs by de-repressing genes such as Pparγ, ultimately reducing the levels of lipid intermediates and preventing lipotoxicity [35,36]. The studies of a HDAC1 inhibitor, tributyrin, and sodium butyrate yielded consistent results. Sodium butyrate improved liver steatosis by upregulating GLP-1R expression and p-AMPK/p-ACC, by inhibiting HDAC2 [37]. Tributyrin inhibited HDAC1, enhanced the CPT1a expression, and improved hepatic steatosis in mice [38]. The reduced HDAC1 activity or HDAC1 knockdown decreased the SREBP1 protein level and attenuated hepatic steatosis (NF-κB/HDAC1/SREBP1c) [39]. The beneficial effect of MS-275 in fatty liver might be mediated by inhibition of HDAC1/2 rather than HDAC3.

Adenosyl phosphoribosyl transferase (APRTase) catalyzes phosphoribosyl transfer from PRPP to adenine, resulting in the formation of AMP and release of pyrophosphate (PPi) via the purine nucleotide salvage pathway [40]. The gene expression levels of APRT and phosphorylation of AMPK were increased in MS-275-treated mice with fatty liver. These results indicate that MS-275 upregulates APRT expression and intracellular AMP, thereby activating AMPK. Because HDAC inhibition is associated with AMPK activation, HDAC1 depletion induced AMPK activation and mitochondrial biogenesis in intestinal epithelial cells [41]. The production of butyrate, an inhibitor of HDAC class I, by polyphenolic derivatives can reduce HDAC1 activity, upregulate AMPK and LC3, and mitigate inflammation in DSS-induced colitis [42].

Because AMPK regulates lipid metabolism, its activity is downregulated in obesity and NAFLD [43,44]. The inhibition of hepatic AMPK activity enables the transition from simple steatosis to NASH [45]. The activation of AMPK reduces de novo lipogenesis by repressing ACC1 and 2, which are CPT-1 inhibitors [46]. The AMPK stimulates lipolysis in hepatocytes, thereby increasing mitochondrial FAO [47]. Consistent with the previous results, MS-275 stimulated the phosphorylation of AMPK, reduced de novo lipogenesis in fatty liver, and increased mitochondrial oxidation in primary hepatocytes. Indeed, hypericin can attenuate nonalcoholic fatty liver disease and abnormal lipid metabolism via PKA-mediated AMPK signaling pathway in vitro and in vivo [48]. The treatment with MS-275 in mice with fatty liver activated the PKA-mediated AMPK signaling pathway and induced the expression of genes related to PGC-1, a mitochondrial biogenesis marker, and TFAM, resulting in the increased rate of mitochondria per unit area.

Fibroblast growth factor 21 (FGF21) is expressed in the liver and is involved in hepatic glucose and lipid metabolism [49]. A long-acting FGF21 drug reduced body weight and improved the lipid profile in an animal model and humans [50]. It also reversed hepatic steatosis by increasing FAO and decreased lipogenesis in diet-induced obese mice, and in mice fed an MCD diet [7,51]. Valproic acid, an inhibitor of HDAC class I, upregulates FGF21 expression in the glia [52]. MS-275 induces hepatic FGF21 expression via H3K18ac-mediated CREBH signaling in primary hepatocytes [53]. We found that several injections of MS-275 significantly increased the FGF21 mRNA and protein levels in the fatty livers of mice, and the secretion of FGF21 into the blood. FGF21 produced by MS-275 stimulation might protect against hepatic steatosis in HFD-induced mice.

In summary, MS-275 has a beneficial effect on hepatic steatosis in HFD mice, as revealed by intravital two-photon imaging. MS-275 restored the dysregulated hepatic lipid metabolism in a HFD-induced NAFLD mouse model. MS-275 stimulated the degradation of hepatic LDs by enhancing lipolysis and mitochondrial FFA oxidation, and reducing de novo lipogenesis. Therefore, MS-275, an inhibitor of HDAC class I, has prophylactic potential for NAFLD.

## 4. Materials and Methods

### 4.1. Animal Studies

All of the animal experiments were approved by the Animal Ethics Committee of Ajou University (number 2021-0005). Six-week-old male C57BL/6J mice were purchased from GEM Pharmatech (Nanjing, China). The mice were housed in a temperature-controlled room at 22 ± 2 °C with a light/dark cycle of 12 h; food pellets were provided ad libitum. After 2 weeks of assimilation, mice were fed a 60% HFD (D12492; Research Diets Inc., New Brunswick, NJ, USA) or normal CD containing 10% fat (D12450B; Research Diets Inc., New Brunswick, NJ, USA) for 14 weeks. The HFD mice were randomly divided into HFD and HFD plus MS-275 groups. Each group was intraperitoneally injected with dimethyl sulfoxide (DMSO) or MS-275 (10 mg/kg), five times every other day. After the DMSO and MS-275 injection, the mice were euthanized for further studies.

### 4.2. Reagents

The Entinostat (MS-275) was purchased from MedChemExpress (Monmouth Junction, NJ, USA). The LD1 and BLT were provided by Prof. Hwan Myung Kim, which showed low cytotoxicity to allow live sample imaging, as confirmed by the previous study [54,55]. The CF405M-conjugated wheat germ agglutinin (WGA) was purchased from Biotium (Fremont, CA, USA). Anti-p-AMPK (#4185S), anti-AMPK (#4150), anti-p-PKA C (#4781S), anti-PKA C-α (#4782), anti-SQSTM1/p62 (#5114S), and anti-LC3B (#2775S) were obtained from Cell Signaling Technology (Danvers, MA, USA). Anti-β-actin (#A300-491A) and anti-FGF21 (#PA5-79254) were purchased from Bethyl Laboratories (Montgomery, TX, USA) and Thermo Fisher Scientific (Waltham, MA, USA), respectively.

### 4.3. Intravital Imaging of the Liver by Two-Photon Microscopy

The LDs and lysosomes were labeled by retro-orbital injection of 150 µL of 100 µM LD-1 and BLT 30 min before intravital imaging. The liver sinusoid endothelial cells were labeled by intravenous injection of CF405M-conjugated WGA (2.5 mg/kg; Biotium) 30 min prior to imaging. The mice were anesthetized with 3–5% isoflurane for induction and 1–2% isoflurane for maintenance. Surgical preparation of the mice for intravital imaging of the liver was performed as described previously [56]. Before surgery, the skin surrounding the abdomen was shaved and the animal was maintained at 37 °C using a commercial heating system (Live Cell Instrument, Seoul, Korea). The abdominal cavity was opened, and the left lateral lobe of the liver was carefully exposed using cotton swabs. The liver lobe was attached to a custom-made silicon pad and covered with a metal frame equipped with a coverslip (22 × 44 mm). To prevent drying of the exposed liver, warm saline (37 °C) was continuously supplied during imaging, which was performed using an upright LSM980 microscope (Carl Zeiss, Oberkochen, Germany) equipped with a MaiTai DeepSee femtosecond-pulsed laser (Spectra-Physics, Santa Clara, CA, USA) at the Three-Dimensional Immune System Imaging Core Facility of Ajou University with an excitation wavelength of 800 nm. The images were acquired using a 20× water immersion objective lens (W Plan-Apochromat 20×; numerical aperture, 1.0) providing a field of view of 212 × 212 μm, with a resolution of 1024 × 1024 pixels and pixel size of 0.2 μm. Z-stack images were acquired from the surface of the liver tissue to the deeper sinusoid bed with hepatocytes (total depth, 30–40 μm), by sequential imaging along the z-axis at 1-μm intervals. The time-lapse imaging was performed at the same location every 2 min for 2–4 h.

### 4.4. Image Data Analysis

ZEN 3.2 (Carl Zeiss), Volocity (Quorum Technologies Inc., Puslinch, ON, Canada), and Imaris 9.3.1 (Bitplane, Zurich, Switzerland) software were used for the 3D image analysis. The motion and volume of hepatic LDs and lysosomes were quantified and visualized using the surface function of Imaris, as previously described [28].

### 4.5. TG Measurements

The TG levels in the tissues were measured using a Triglyceride Quantification Colorimetric/Fluorometric Kit (BioVision, Milpitas, CA, USA), according to the manufacturer’s instructions. Briefly, tissues were lysed in 5% NP40 solution, homogenized, and boiled for 5 min. After centrifugation (10,000× *g*, 2 min), 5 μL of supernatant was diluted with 130 μL of TG assay buffer to prepare TG solutions. Next, 50 μL of each TG solution was digested with 2 μL of lipase at room temperature (RT) for 20 min. The digested TG solution was mixed with 50 μL of TG reaction mix. After incubation at RT for 60 min, absorbance at 570 nm was measured using a microplate reader (Bio-Rad, Hercules, CA, USA). The TG levels were determined using a standard curve.

### 4.6. Histological Analysis

The liver tissues were fixed with phosphate-buffered saline (PBS) containing 4% paraformaldehyde, embedded in paraffin, sliced into 5-μm-thick sections, mounted onto slides, and sequentially stained with H&E (Abcam, Cambridge, UK), according to standard procedures. For LD staining, the liver tissues were frozen in Optimum Cutting Temperature compound (OCT; Leica Biosystems Richmond Inc., Richmond, IL, USA). Sections (5 μm) were fixed in 10% formalin for 30 min, soaked in 60% isopropanol for 5 min, and stained with ORO (0.5% in 60% isopropanol) for 10 min. After rinsing sections three times with water, the nuclei were stained by soaking in hematoxylin solution for 1 min. The slides were observed with an Aperio ScanScope CS Slide Scanner (Leica, Wetzlar, Germany).

### 4.7. AST and ALT Measurements

The blood obtained from the mouse heart was immediately centrifuged (1000× *g*, 10 min) at 4 °C. The upper plasma layer was collected and stored at −80 °C. The plasma AST and ALT levels were measured using ASAT/GOT and ALAT/GPT Kits (Roche Diagnostics, Mannheim, Germany), according to the manufacturer’s instructions. Briefly, AST and ALT produce oxaloacetate and pyruvate from aspartate and alanine, respectively. The nicotinamide adenine dinucleotide hydrogen (NADH) reduction during the conversion of oxaloacetate to malate and pyruvate to lactate, by malic enzyme and lactate dehydrogenase, respectively, was quantified using an autochemical analyzer (7600; Hitachi, Tokyo, Japan). The NADH reduction was proportional to the activities of AST and ALT.

### 4.8. Transmission Electron Microscopy

The mouse liver tissues were fixed with Karnovsky’s fixative solution (1% paraformaldehyde, 2% glutaraldehyde, 2 mM calcium chloride, and 100 mM cacodylate buffer (pH 7.4)) for 2 h and washed with cacodylate buffer. After postfixing with fixative solution containing 1% osmium tetroxide and 1.5% potassium ferrocyanide for 1 h, the tissues were dehydrated with 50–100% alcohol, stained en bloc in 0.5% uranyl acetate, embedded in Poly/Bed^®^ 812 resin (Ted Pella Inc., Redding, CA, USA) polymerized, sectioned using a Reichert Jung Ultracut S (Leica), and stained with uranyl acetate and lead citrate. The mouse liver tissues were observed by TEM (EM 902A; Carl Zeiss MicroImaging GmbH, Göttingen, Germany).

### 4.9. Isolation of Mouse primary Hepatocytes

The primary hepatocytes were isolated from 8-week-old male C57BL/6J mice. Briefly, mice were anesthetized with isoflurane and their livers were perfused with pre-perfusion buffer (140 mmol/L NaCl, 6 mmol/L KCl, 10 mmol/L HEPES, and 0.2 mmol/L EGTA (pH 7.4)) at 7 mL/min for 5 min, followed by continuous perfusion with collagenase-containing buffer (66.7 mmol/L NaCl, 6.7 mmol/L KCl, 5 mmol/L HEPES, 0.48 mmol/L CaCl_2_, and 0.8 mg/mL type-IV collagenase (pH 7.4)) for 8 min. Viable hepatocytes were harvested by centrifugation (50× *g*, 5 min) and suspended in Media 199 (Invitrogen, Carlsbad, CA, USA). The hepatocyte suspension was layered on a 40% Percoll cushion and centrifuged (300× *g*, 7 min). The pellets were collected and resuspended in Media 199 (Invitrogen). Hepatocyte viability was evaluated by trypan blue exclusion assay. A viability of ≥85% was required for the hepatocytes used in this study. The hepatocytes were resuspended in complete growth medium (medium 199 containing 10% fetal bovine serum (FBS)) and seeded into collagen-coated 96-well plates at a density of 2 × 10^4^/well.

### 4.10. Measurement of FAO Rate

The FAO rate was determined using a Fatty Acid Oxidation Assay Kit (ab217602; Abcam) and Extracellular O_2_ Consumption Assay Kit (ab197243; Abcam). Briefly, the cells were seeded in 96-well culture plates and cultured in glucose-free RPMI 1640 medium at 37 °C overnight. The culture medium was removed and replaced with 150 μL of assay medium containing oleate as the FAO substrate, followed by incubation at 37 °C for 30 min in a CO_2_ incubator. The FAO rate was determined by measuring extracellular O_2_ consumption (EOC) in assay medium. EOC was measured based on the ability of oxygen to quench the excited state of the kit reagent, which was added in a 10-μL amount (along with two drops of pre-warmed high-sensitivity mineral oil) to each well containing assay medium. The emitted fluorescence intensity at 650 nm after excitation at 380 nm was measured over 80 min using a plate reader (Molecular Devices, San Jose, CA, USA). The fluorescence values were summed. The FAO rate was determined by comparing MS-275- and DMSO-treated MPHs.

### 4.11. Western Blot Analysis

Mouse liver tissues were lysed with radioimmunoprecipitation (RIPA) buffer (150 mM NaCl, 1% NP-40, 0.5% deoxycholate, 0.1% sodium dodecyl sulfate (SDS), and 50 mM Tris-HCl (pH 7.5)) supplemented with protease inhibitor cocktail. Equal concentrations of proteins were diluted in the SDS sample buffer (50 mM Tris-Cl at pH 6.8, 2% SDS, 100 mM DL-dithiothreitol (DTT), 10% glycerol), separated on an 8–12% polyacrylamide gel, and transferred to a polyvinylidene fluoride (PVDF) membrane. After blocking the membrane with 5% skim milk for 30 min, the target antigen was reacted with the primary antibody at RT for 2 h. The membrane was then incubated with a horseradish peroxidase-conjugated anti-mouse or anti-rabbit IgG secondary antibody at RT for 1 h. The immunoreactive bands were detected using an enhanced chemiluminescence system (Pierce ECL Western Blotting Substrate; Thermo Fisher Scientific). The band intensity was measured using Quantity One 1D image analysis software (Bio-Rad).

### 4.12. Quantitative Reverse Transcriptase-Polymerase Chain Reaction (qRT-PCR)

Total RNA was extracted from the liver tissues with RNAiso Plus reagent (TaKaRa Bio, Shiga, Japan). The cDNAs were synthesized using the AMV reverse transcriptase and random 9-mers in the TaKaRa RNA PCR Kit (version 3.0; TaKaRa Bio). The primer sets used for PCR amplification are listed in Appendix A. Quantitative reverse-transcription PCR was performed with SYBR Green (TaKaRa), using a TaKaRa TP-815 instrument. The relative quantities of amplified DNA were analyzed using the software bundled with the TP-815 instrument and normalized to the mouse 36B4 mRNA level.

### 4.13. Statistical Analysis

The experiments were repeated at least three times. The data are means ± standard error of the mean (SEM), and were analyzed using Prism 9.0 software (GraphPad Software Inc., San Diego, CA, USA). For the statistical analysis, one-way analysis of variance (ANOVA) with the Bonferroni post-hoc test was used. A *p*-value < 0.05 was considered statistically significant.

## 5. Conclusions

We evaluated the effect of MS-275 on the degradation of hepatic LDs and FFA oxidation in HFD-induced fatty liver. Short-term treatment with MS-275 improved the hepatic metabolic alterations in HFD-induced fatty liver. Intravital two-photon imaging analysis demonstrated that MS-275 increased the degradation of hepatic LDs, and promoted interactions of LDs with lysozymes in fatty liver. PDM, lipolysis, and lipophagy were observed in the MS-275-treated mouse fatty liver. The lipolysis and autophagy pathways were activated in the livers of the MS-275 treated mice. In addition, MS-275 reduced de novo lipogenesis, but increased the expression of mitochondrial oxidation and oxidation-related genes, such as *Pparα*, *Acadm*, *Cpt1b*, and *Fgf21*. Taken together, the results show that MS-275 stimulates the degradation of hepatic LDs and enhances mitochondrial FFA oxidation, thus protecting against HFD-induced NAFLD in C57BL/6J mice. Therefore, MS-275 has prophylactic and therapeutic potential for NAFLD and NASH.

## Figures and Tables

**Figure 1 ijms-23-09978-f001:**
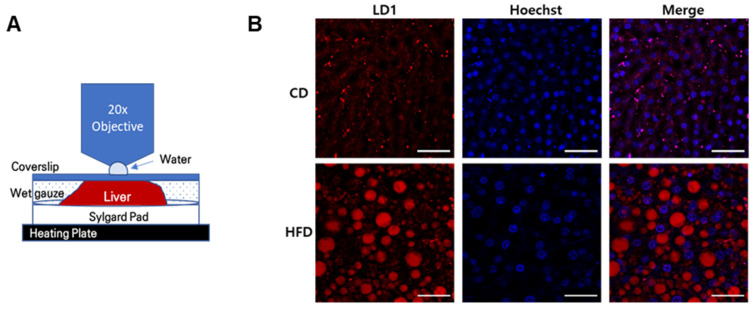
Intravital two-photon imaging of hepatic lipid droplet (LD) accumulation in a high-fat diet (HFD)-induced fatty liver mouse model. C57BL/6J mice were fed a chow diet (CD) or HFD for 14 weeks. Hepatic LDs in mice were stained by intravenous injection with LD1 fluorescent dye (red) and detected by intravital two-photon microscopy. (**A**) Scheme of liver intravital imaging procedure; (**B**) The 3D rendering of intravital two-photon microscopy images of LD accumulation. The nucleus was stained with Hoechst (blue) dye. Scale bar = 50 μm (**C**) LD accumulation was quantified based on LD volume of total image spot. *** *p* < 0.001 (one-way ANOVA); (**D**) LD volumes were classified as <1000, 1000–10,000, or >10,000 µm^3^; (**E**) Snapshot images showing the fusion of LDs (yellow arrows) taken from timelapse Appendix A; Scale bar = 5 μm. (**F**) Immunoblots of Class 1 HDAC protein levels (HDAC1–3) in liver tissues of C57BL/6J mice fed a HFD for 14 weeks. ** *p* < 0.01 (unpaired *t*-test).

**Figure 2 ijms-23-09978-f002:**
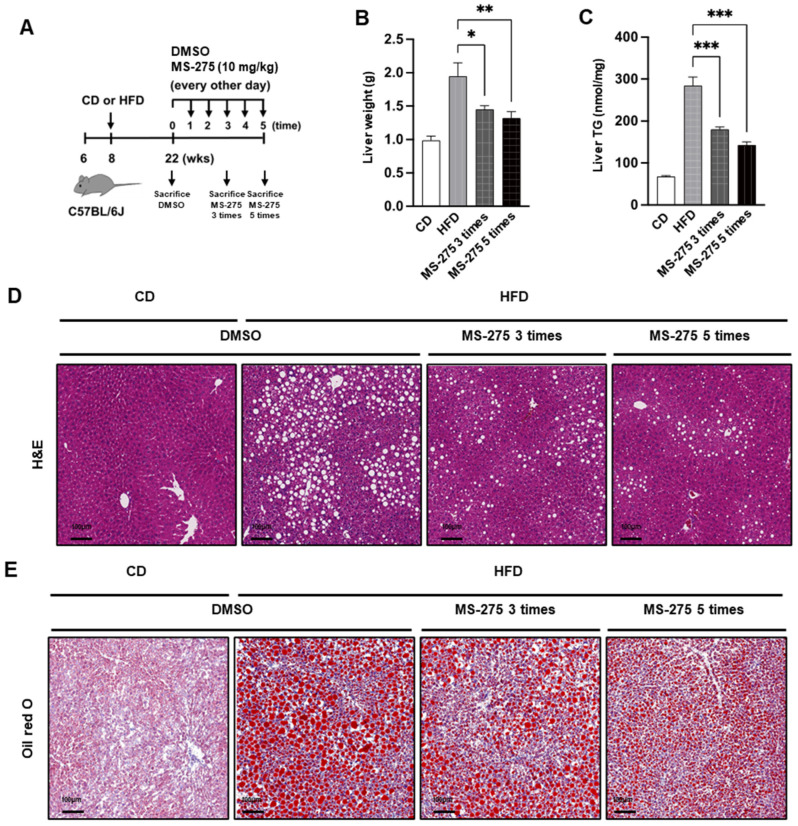
MS-275 ameliorated HFD-induced fatty liver in C57BL/6J mice. C57BL/6J mice were fed a CD or HFD for 14 weeks and intraperitoneally injected with dimethyl sulfoxide (DMSO) or MS-275 (10 mg/kg), five times every other day (*n* = 5). (**A**) Schematic procedure of animal experiments; (**B**) Liver weight and (**C**) Hepatic TG content were analyzed at the end of the experiment. Hepatic lipids were detected by (**D**) H&E and (**E**) oil red O staining; Scale bar = 100 μm. (**F**) Plasma levels of triacylglycerol (TG), total cholesterol (TCHO), aspartate aminotransferase (AST), and alanine aminotransferase (ALT) were measured using an auto-chemical analyzer. * *p* < 0.05, ** *p* < 0.01, *** *p* < 0.001 vs. HFD (one-way ANOVA).

**Figure 3 ijms-23-09978-f003:**
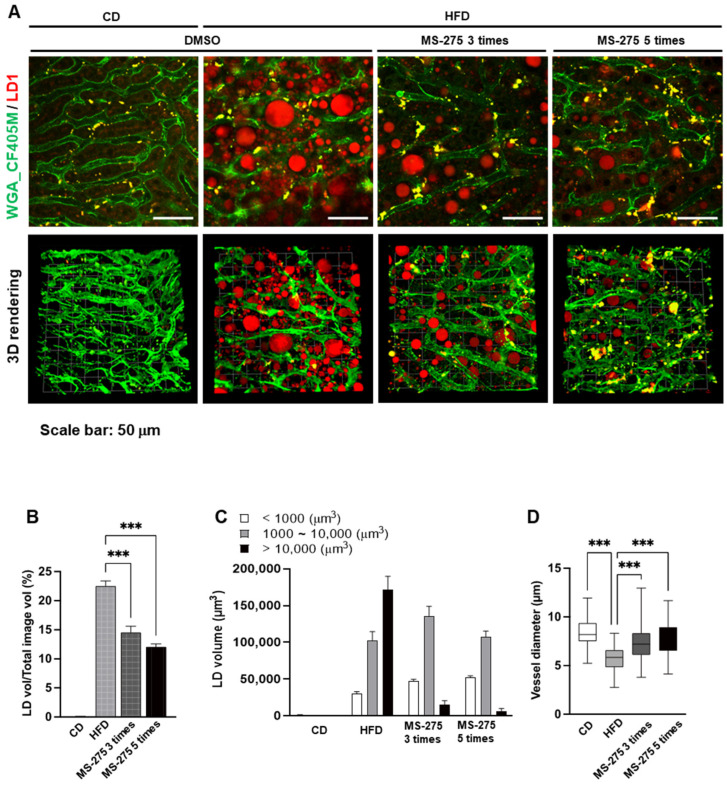
MS-275 stimulated degradation of hepatic LDs in HFD-induced fatty liver. DMSO or MS-275 treated mice were intravenously injected with WGA antibody (green) to stain blood vessels, and LD1 (red) to stain LDs. (**A**) Hepatic LDs and blood vessels were visualized by intravital two-photon imaging and 3D rendering using a two-photon microscope in 0, 3, and 5 times of MS-275 injection; (**B**) LD accumulation was quantified based on the LD volume; (**C**) LD volumes were classified as <1000, 1000–10,000, or >10,000 µm^3^; (**D**) Hepatic sinusoidal vessels were quantified based on vessel diameter. *** *p* < 0.001 (one-way ANOVA).

**Figure 4 ijms-23-09978-f004:**
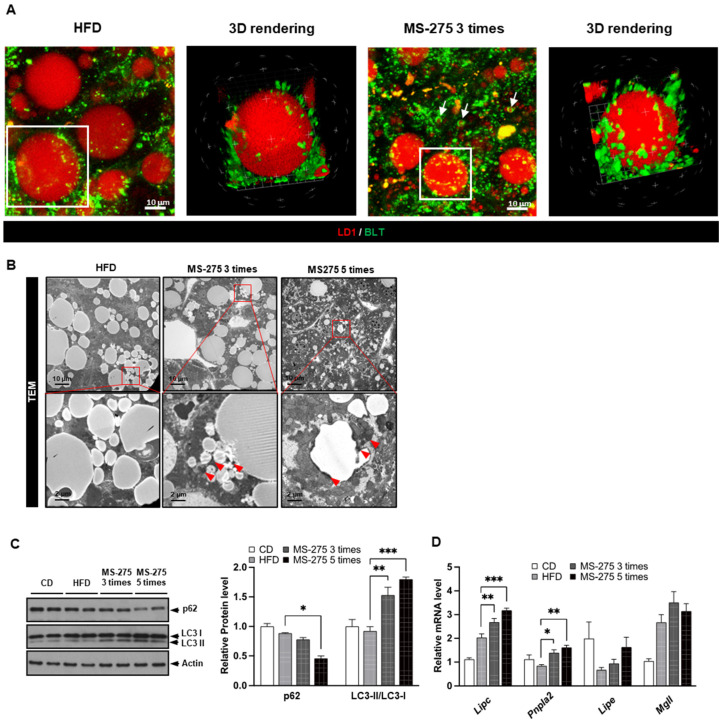
MS-275 induced lipophagy and increased lipolysis-related gene expression in HFD-induced fatty liver. (**A**) Co-localization of LDs and lysosomes in liver tissue from MS-275-treated mice was visualized by two-photon microscopy staining with LD1 (red) for LDs and BLT (green) for lysosomes. Scaled-up and z-stack image represents LDs surrounded by lysosomes; (**B**) Cellular organelles in liver tissues of MS-275-treated mice were visualized by transmission electron microscopy (TEM). Red arrows indicates lipophagy; (**C**) Conversion of LC3I into LC3II and p62 degradation in liver tissues of MS-275-treated mice was detected by immunoblotting; (**D**) Expression levels of genes related to lipolysis (Lipc, Pnpla2, Lipe, and Mgll) were determined by quantitative reverse transcriptase-polymerase chain reaction (qRT-PCR); (**E**) Expression levels of Aprt were determined by qRT-PCR; (**F**) p-AMPK and p-AKT levels in liver tissues of MS-275-treated mice were determined by immunoblotting. * *p* < 0.05, ** *p* < 0.01 and *** *p* < 0.001 (one-way ANOVA).

**Figure 5 ijms-23-09978-f005:**
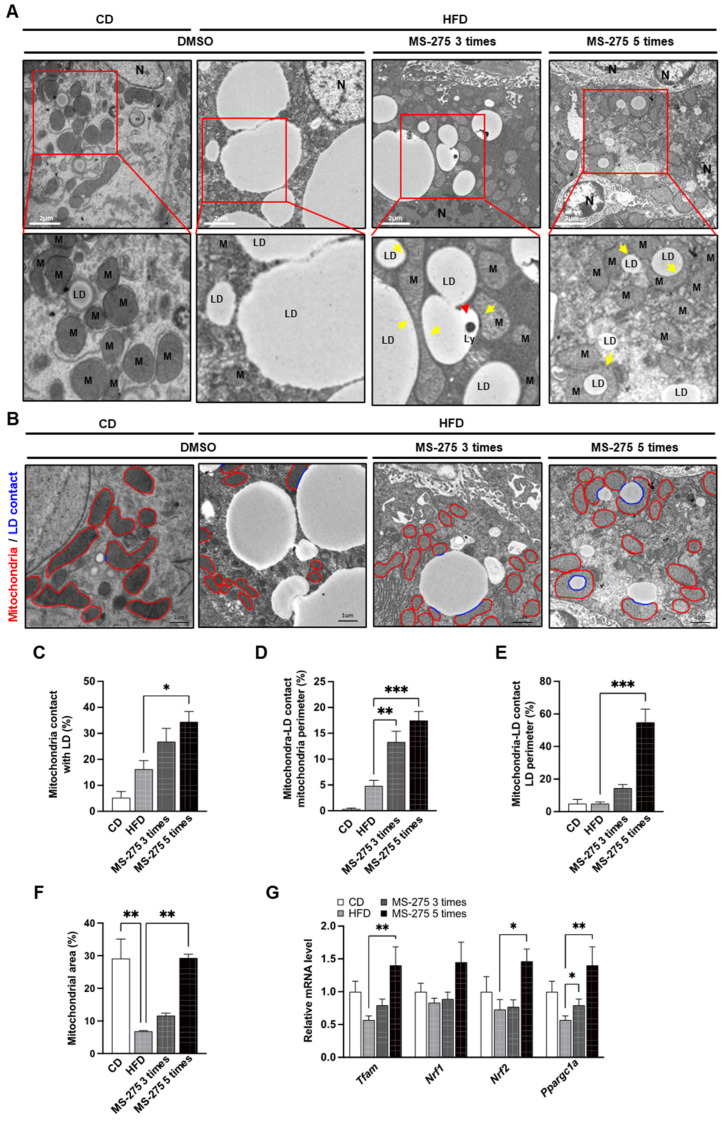
MS-275 increased mitochondrial content and the expression of mitochondrial biogenesis-related genes, as well as peri-lipid-droplet mitochondria in HFD-induced fatty liver. (**A**) Liver tissues isolated from mice administered MS-275 three or five times after feeding a CD or HFD for 14 weeks were observed by TEM (N, nuclei; LD, lipid droplet; M, mitochondria; Ly, lysosome; red arrow, lysosome; yellow arrow, peri-droplet mitochondria (PDM); (**B**) Cytosolic mitochondria and PDM contents, as revealed by TEM. Red line represents boundary of mitochondria and blue line represents LDs in contact with mitochondria; (**C**) PDM abundance; (**D**) Mitochondrial interaction with LDs; (**E**) LD interactions with mitochondria; (**F**) Mitochondrial area fraction relative to total cytosol; (**G**) The expression levels of genes related to mitochondrial biogenesis (Tfam, Nrf, and Ppargc1a) were determined by qRT-PCR. * *p* < 0.05, ** *p* < 0.01 and *** *p* < 0.001 (one-way ANOVA).

**Figure 6 ijms-23-09978-f006:**
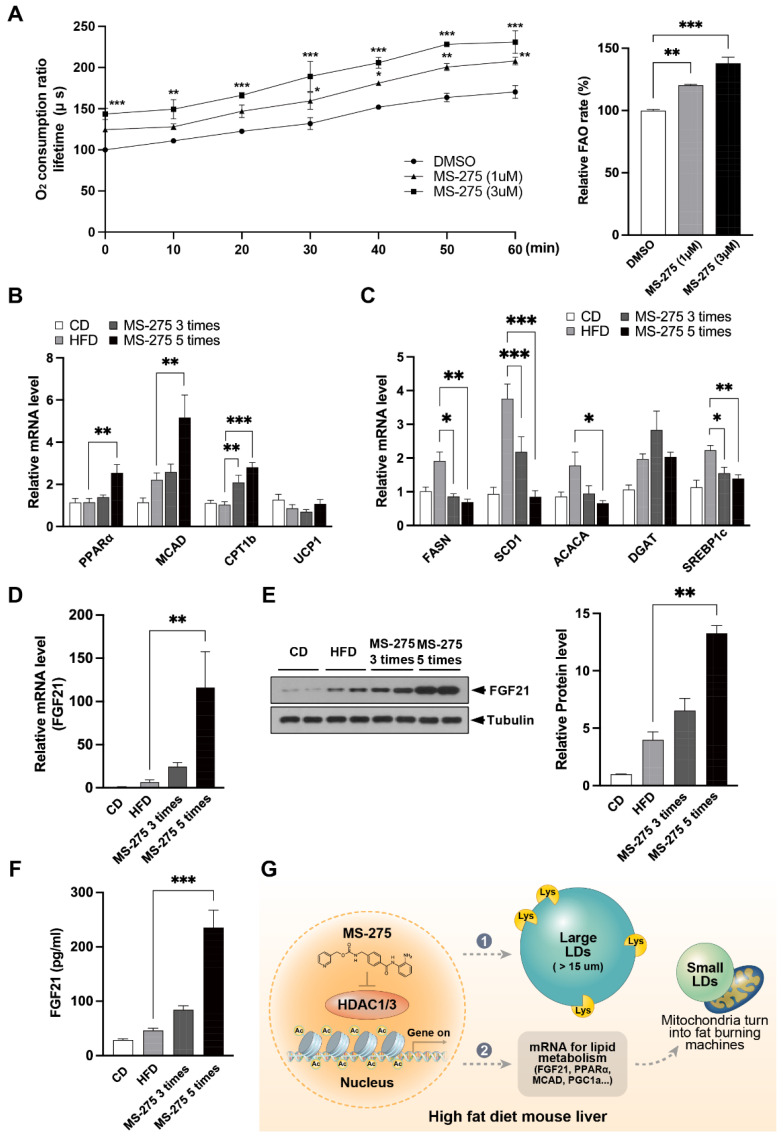
MS-275 increased mitochondrial oxidation in mouse primary hepatocytes, and the expression of FGF21 in HFD-induced fatty liver. (**A**) Mouse primary hepatocytes were isolated from 8-week-old C57BL/6J mice. Mitochondrial respiration in mouse primary hepatocytes exposed to 1 or 3 µM MS-275 was investigated by measuring the mitochondrial oxygen consumption rate (OCR); (**B**,**C**) mRNA expression levels of FAO-related genes (PPARα, Acadm, Cpt1b, and Ucp1) and lipid synthesis-related genes (Fasn, Dgat, Scd, Acaca, Srebp1c) in liver tissue injected with MS-275 were determined by qRT-PCR; (**D**) Expression of FGF21 was determined by qRT-PCR; (**E**) FGF21 levels in liver tissue injected with MS-275, as determined by immunoblotting; (**F**) Plasma levels of FGF21 measured using an FGF21 assay kit; (**G**) Mechanism of HDAC1/3 inhibition of HFD-induced steatotic liver. * *p* < 0.05, ** *p* < 0.01, *** *p* < 0.001 (one-way ANOVA).

## Data Availability

The data presented in this study are available on request from the corresponding author (J.Y.J.; twinstwins@hanmail.net and J.Y.K.; jykwak@aumc.ac.kr).

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
