# Peer review of "In Vivo Two-Photon Imaging Analysis of Dynamic Degradation of Hepatic Lipid Droplets in MS-275-Treated Mouse Liver"

_ijms, 2022, doi:10.3390/ijms23179978_

Round 1

Reviewer 1 Report

Lee et al. have shown that MS-275 significantly protected from the high-fat diet-induced fatty liver via promoting lipolysis. The mouse data about the MS-275 treatment are exciting and impressive. The proposed molecular mechanism is not convincing due to the lack of cell culture experiments. However, in situ imaging of the mouse liver provides direct evidence that supports the conclusion. The main issue is on the results of figure 3: (1) the timepoint when these images were taken should be described in results and legends to evaluate the efficacy of the MS-275 treatment; (2)because the dynamic degradation was recorded for 2-4 hours, authors should be able to provide a video showing increased lipolysis in MS-275-treated liver compared to DMSO-treated liver. This article is suitable to publish If the authors can provide these critical data. 

Author Response

August 23, 2022

To: Editor-in-Chief International Journal of Molecular Sciences

CC: Guest Editor, Special Issue “Molecular Imaging in Nanomedical Research 3.0”

We are resubmitting our revised, research manuscript entitled “In Vivo Two-photon Imaging Analysis of Dynamic Degradation of Hepatic Lipid Droplets in MS-275-Treated Mouse Liver.” by Lee et. al. for reconsideration for publication in International Journal of Molecular Sciences.

We received the review report (minor revision) on August 18, 2022. The paper was reviewed by two excellent reviewers who provided constructive critiques. We responded to each reviewer by point-by-point manner (see response to reviewers below). Additionally, we sent out the original manuscript for English proofreading by a professional English editing service and received the proofread manuscript on July 27, 2022 (certificate uploaded on submission site).

The revised manuscript is original and has not been submitted for publication elsewhere. All authors have read the revised manuscript and approved its resubmission. There are no known conflicts of interest associated with this publication and there has been no significant financial support for this study that could have influenced its outcome. The approval of relevant bodies and such approvals are acknowledged within the manuscript. Thank you for considering this manuscript. We look forward to your reply.

Sincerely,

Ja Young Jeon, MD, Ph.D.

Department of Endocrinology and Metabolism, Ajou University School of Medicine, 164, World cup-ro, Yeongtong-gu, Suwon, 16499, Republic of Korea, Tel: +82 31 219 7459, Fax: +82 31 219 4497, E-mail: twinstwins@hanmail.net

Reviewer 1

Lee et al. have shown that MS-275 significantly protected from the high-fat diet-induced fatty liver via promoting lipolysis. The mouse data about the MS-275 treatment are exciting and impressive. The proposed molecular mechanism is not convincing due to the lack of cell culture experiments. However, in situ imaging of the mouse liver provides direct evidence that supports the conclusion.

Reviewer 1, comment 1

The main issue is on the results of figure 3:

(1) the timepoint when these images were taken should be described in results and legends to evaluate the efficacy of the MS-275 treatment;

Response to Reviewer 1, comment 1

All in vivo data follows the scheme of Figure 2A. After 14 weeks of HFD feeding, MS-275 was injected for every other day. We analyzed at the timepoint of 0, 3 and 5 times of injection to evaluate the efficacy of the MS-275 treatment. As per the reviewer’s comments, we described in results and legends about the timepoint when these images were taken.

  • Line 172-175: As shown in Figure 3A, MS-275-treated mouse liver had smaller LDs than the HFD-fed mouse liver (Figure 3A). The volumes of LDs were significantly decreased by the number of injection times with MS-275.
  • Line 187-190: Hepatic LDs and blood vessels were visualized by intravital two-photon imaging and 3D rendering using a two-photon microscope in 0, 3 and 5 times of MS-275 injection.

Reviewer 1, comment 2

(2) because the dynamic degradation was recorded for 2-4 hours, authors should be able to provide a video showing increased lipolysis in MS-275-treated liver compared to DMSO-treated liver. This article is suitable to publish If the authors can provide these critical data. 

Response to Reviewer 1, comment 2

We substituted Supplementary Video S3 to compare increased lipolysis in MS-275-treated liver compared to DMSO-treated liver as per the reviewer’s recommendation.

Reviewer 2 Report

The study by Dr Lee and team reports a HDAC1/3 inhibitor MS-275 induced liver LD dynamic degradation observed under intravital two photon microscopy. Although the drug MS-275 has been well studied in NAFLD/NASH, its role in lipophagy and influence on the dynamic changes of LDs not yet well understood. With fluorescence labelling of LDs, the authors studied the dynamic alterations and interactions of LDs with lysosomes etc, and further examined the involvement of mitochondria and certain molecular pathways for LD degradation in steatosis. The observations are interesting and the manuscript is well written, yet, some comments are provided for the authors to address:

Major

1.     How were LD volumes accurately calculated as the imaging was only at one plane but the location of LDs is 3-dimensional? Is it more reasonable to present the data in LD area or diameters? Even though, the authors should state clearly the criteria of LD selection for quantification. 

2.     Introduction lines 88-89, the cited reference 18 did not mention LDs at all, instead, the in vivo labeling and multiphoton intravital imaging of LDs in HFD-induced NAFLD liver has been recently reported by Wang et al. in Advanced Materials (PMID: 33576084) but neglected here. Other references should be carefully checked too.

3.     In addition, if the florescence dye LD1 used for LDs labelling toxic and its possible influence on LD dynamics, thus confounding the observed phenomenon, should be discussed.

4.     Fig 2, The presentation of experimental design is confusing, is it 5 times/injects every other day or total injection 5 times? It is also confusing what is 3 times or 5 times.

5.     The authors report that a HDAC inhibitor MS-275 attenuates liver steatosis in a HFD NALFD animal model. The authors propose the involvement of lysosomes and mitochondria pathways in this pathological process and LD alterations. Since ceramides have been recently noticed highly associated with lipogenesis and steatosis at almost every step in NAFLD (PMID: 35777449), it would be interesting to know if MS-275 has any impact on ceramide production in this study. Alternatively, the authors may elaborate the potential involvement of ceramides in the observed phenomenon in the Discussion.

Minor

6.     Language editing/checking might be necessary, e.g. in Fig 6, it should be “mRNA”

Author Response

August 23, 2022

To: Editor-in-Chief International Journal of Molecular Sciences

CC: Guest Editor, Special Issue “Molecular Imaging in Nanomedical Research 3.0”

We are resubmitting our revised, research manuscript entitled “In Vivo Two-photon Imaging Analysis of Dynamic Degradation of Hepatic Lipid Droplets in MS-275-Treated Mouse Liver.” by Lee et. al. for reconsideration for publication in International Journal of Molecular Sciences.

We received the review report (minor revision) on August 18, 2022. The paper was reviewed by two excellent reviewers who provided constructive critiques. We responded to each reviewer by point-by-point manner (see response to reviewers below). Additionally, we sent out the original manuscript for English proofreading by a professional English editing service and received the proofread manuscript on July 27, 2022 (certificate uploaded on submission site).

The revised manuscript is original and has not been submitted for publication elsewhere. All authors have read the revised manuscript and approved its resubmission. There are no known conflicts of interest associated with this publication and there has been no significant financial support for this study that could have influenced its outcome. The approval of relevant bodies and such approvals are acknowledged within the manuscript. Thank you for considering this manuscript. We look forward to your reply.

Sincerely,

Ja Young Jeon, MD, Ph.D.

Department of Endocrinology and Metabolism, Ajou University School of Medicine, 164, World cup-ro, Yeongtong-gu, Suwon, 16499, Republic of Korea, Tel: +82 31 219 7459, Fax: +82 31 219 4497, E-mail: twinstwins@hanmail.net

Reviewer 2

The study by Dr Lee and team reports a HDAC1/3 inhibitor MS-275 induced liver LD dynamic degradation observed under intravital two photon microscopy. Although the drug MS-275 has been well studied in NAFLD/NASH, its role in lipophagy and influence on the dynamic changes of LDs not yet well understood. With fluorescence labelling of LDs, the authors studied the dynamic alterations and interactions of LDs with lysosomes etc, and further examined the involvement of mitochondria and certain molecular pathways for LD degradation in steatosis. The observations are interesting and the manuscript is well written, yet, some comments are provided for the authors to address:

Reviewer 2, comment 1

Major

  1. How were LD volumes accurately calculated as the imaging was only at one plane but the location of LDs is 3-dimensional? Is it more reasonable to present the data in LD area or diameters? Even though, the authors should state clearly the criteria of LD selection for quantification. 

Response to Reviewer 2, comment 1

We used z-stack to take x, y, z axis of image and rendered using ‘IMARIS’ software (Bitplane, Zurich, Switzerland) to 3D image. The ‘Surface’ function in IMARIS software can automatically measure the LDs as previously described in cited reference 27. We supplemented reference 27 in the materials and methods section (4.4.) to describe about the quantification of hepatic LD volume measurement.

  • 4. Image data analysis

ZEN 3.2 (Carl Zeiss), Volocity (Quorum Technologies Inc., Puslinch, Canada), and Imaris 9.3.1 (Bitplane, Zurich, Switzerland) software were used for 3D image analysis. The motion and volume of hepatic LDs and lysosomes were quantified and visualized using the surface function of Imaris as previously described [28].

  1. Moon, J.; Jeon, J.; Kong, E.; Hong, S.; Lee, J.; Lee, E.K.; Kim, P. Intravital two-photon imaging and quantification of hepatic steatosis and fibrosis in a live small animal model. Biomed Opt Express 2021, 12, 7918-7927, doi:10.1364/BOE.442608.

Reviewer 2, comment 2

  1. Introduction lines 88-89, the cited reference 18 did not mention LDs at all, instead, the in vivo labeling and multiphoton intravital imaging of LDs in HFD-induced NAFLD liver has been recently reported by Wang et al. in Advanced Materials (PMID: 33576084) but neglected here. Other references should be carefully checked too.

Response to Reviewer 2, comment 2

We agree that Wang et al. (PMID: 33576084) is well described about in vivo labeling and multiphoton intravital imaging of LDs in HFD-induced NAFLD liver. We substituted reference 18 into suggested references. In addition, we have checked other references were also properly cited too.

  1. Wang, S.; Li, X.; Chong, S.Y.; Wang, X.; Chen, H.; Chen, C.; Ng, L.G.; Wang, J.-W.; Liu, B. In Vivo Three-Photon Imaging of Lipids using Ultrabright Fluorogens with Aggregation-Induced Emission. Advanced Materials 2021, 33, 2007490, doi:https://doi.org/10.1002/adma.202007490.

Reviewer 2, comment 3

  1. In addition, if the florescence dye LD1 used for LDs labelling toxic and its possible influence on LD dynamics, thus confounding the observed phenomenon, should be discussed.

Response to Reviewer 2, comment 3

We really appreciate for the reviewer’s valuable comments. Indeed, LDs dynamics are sensitive, which might be perturbed by the exogenous fluorescent dyes. It is reported that small molecular size, high target specificity, low concentration of the fluorescent dyes could minimize such perturbation (Molecules 2022, 27, 4501; Fungal Biol. Rev. 2022, 41, 45; ACS Cent. Sci. 2021, 7, 1561; Chem. Soc. Rev., 2015, 44, 4953; Angew. Chem. Int. Ed. 2009, 48, 1498). In this regard, we used small, lipophilic, and ultra-high LDs-specific dye (LD1) optimized in micromolar range and short incubation time. Further, LD1 also exhibited low cytotoxicity to allow live sample imaging as confirmed by the previous study (ACS Sens. 2022, 7, 1027). We supplemented additional information about the low cytotoxicity of LD1 in the Materials and Methods (4.2.) section, including reference 54 and 55.

  • Line 395-396: LD1 and BLT were provided by Prof. Hwan Myung Kim, which exhibited low cytotoxicity to allow live sample imaging as confirmed by the previous study [54,55].
  1. Han, J.H.; Park, S.K.; Lim, C.S.; Park, M.K.; Kim, H.J.; Kim, H.M.; Cho, B.R. Simultaneous imaging of mitochondria and lysosomes by using two-photon fluorescent probes. Chemistry 2012, 18, 15246-15249, doi:10.1002/chem.201203452.
  2. Lee, H.W.; Lee, I.J.; Lee, S.J.; Kim, Y.R.; Kim, H.M. Highly Sensitive Two-Photon Lipid Droplet Tracker for In Vivo Screening of Drug Induced Liver Injury. ACS Sens 2022, 7, 1027-1035, doi:10.1021/acssensors.1c02679.

Reviewer 2, comment 4

  1. Fig 2, The presentation of experimental design is confusing, is it 5 times/injects every other day or total injection 5 times? It is also confusing what is 3 times or 5 times.

Response to Reviewer 2, comment 4

We injected MS-275 or vehicle (DMSO) by every other day, for a total of 5 injections when the mice were 22 weeks. To clarify the process of the experiment, we changed to new scheme of experiment in Fig 2A as per the reviewer’s comments.

  • Fig 2A.

Reviewer 2, comment 5

  1. The authors report that a HDAC inhibitor MS-275 attenuates liver steatosis in a HFD NALFD animal model. The authors propose the involvement of lysosomes and mitochondria pathways in this pathological process and LD alterations. Since ceramides have been recently noticed highly associated with lipogenesis and steatosis at almost every step in NAFLD (PMID: 35777449), it would be interesting to know if MS-275 has any impact on ceramide production in this study. Alternatively, the authors may elaborate the potential involvement of ceramides in the observed phenomenon in the Discussion.

Response to Reviewer 2, comment 5

We agree that ceramide is highly associated with lipogenesis, impaired mitochondrial fatty acid oxidation and steatosis at almost every step in NAFLD, which might be predicted that MS-275 prevent NAFLD by inhibiting ceramide production in our study. As per the reviewer’s recommendation, we supplied additional statements about the potential involvement of ceramides in the observed phenomenon in the Discussion section.

  • Line 312-321: Recent studies have been revealed that ceramide contributes to de novo lipogenesis, pathogenesis of NAFLD and impaired mitochondrial fatty acid oxidation [29, 30]. In our study, treatment of MS-275 decreased mRNA expression levels of fatty acid synthase, which is known to regulate palmitoyl-CoA expression, and it might be predicted to inhibit ceramide production [31]. Therefore, it would be considered that inhibiting ceramide production and toxicity by MS-275 might prevent NAFLD. However, treat-ment with MS-275 had no significant effect on ceramide generation in cancer cell [32], more detailed further investigation is required to determine whether MS-275 affects the ceramide production in metabolic disease states such as NAFLD.
  1. Yu, X.D.; Wang, J.W. Ceramide de novo synthesis in non-alcoholic fatty liver disease: Pathogenic mechanisms and therapeutic perspectives. Biochem. Pharmacol. 2022, 202, 115157, doi:10.1016/j.bcp.2022.115157.
  2. Fucho, R.; Casals, N.; Serra, D.; Herrero, L. Ceramides and mitochondrial fatty acid oxidation in obesity. FASEB J. 2017, 31, 1263-1272, doi:10.1096/fj.201601156R.
  3. Bikman, B.T.; Summers, S.A. Ceramides as modulators of cellular and whole-body metabolism. J. Clin. Invest. 2011, 121, 4222-4230, doi:10.1172/jci57144.
  4. Meyers-Needham, M.; Ponnusamy, S.; Gencer, S.; Jiang, W.; Thomas, R.J.; Senkal, C.E.; Ogretmen, B. Concerted functions of HDAC1 and microRNA-574-5p repress alternatively spliced ceramide synthase 1 expression in human cancer cells. EMBO Mol. Med. 2012, 4, 78-92, doi:10.1002/emmm.201100189.

Reviewer 2, comment 6

Minor

  1. Language editing/checking might be necessary, e.g. in Fig 6, it should be “mRNA”

Response to Reviewer 2, comment 6

We have corrected Fig 3 and Fig 6 and checked throughout the manuscript about the language as per the reviewer’s comments.

  • Fig 3A. “LD5” -> “LD1”
  • Fig 6G. “mRnA” -> “mRNA”
